# Misclassification excess risk bounds for PAC-Bayesian classification via convexified loss

## Abstract

PAC-Bayesian bounds have proven to be a valuable tool for deriving generalization bounds and for designing new learning algorithms in machine learning. However, they typically focus on providing generalization bounds with respect to a chosen loss function. In this study, we concentrate on the problem of PAC-Bayesian classification, specifically referring to the PAC-Bayesian method for binary classification. In classification tasks, due to the non-convex nature of the 0-1 loss, a convex surrogate loss is often used, and thus current PAC-Bayesian bounds are primarily specified for this convex surrogate. This work introduce a procedure to obtain misclassification excess risk bounds for PAC-Bayesian classification when using a convex surrogate loss. Our key ingredient here is to leverage PAC-Bayesian relative bounds in expectation rather than relying on PAC-Bayesian bounds in probability. We demonstrate our approach in several important applications: high-dimensional sparse classification and 1-bit matrix completion.

Keyword: binary classification, PAC-Bayes bounds, prediction bounds, misclassification excess risk, convex surrogate loss

## 1 Introduction and motivation

Building on foundational works by Shawe-Taylor and Williamson (1997); McAllester (1998; 1999), PAC-Bayesian theory has emerged as a vital framework for deriving generalization bounds and developing innovative learning algorithms in machine learning (Catoni, 2007; Guedj, 2019; Alquier, 2024; Rivasplata, 2022; Germain et al., 2009; Reeb et al., 2018). PAC-Bayesian bounds typically focus on risk assessments related to specific loss functions. However, in classification, the non-convex and non-smooth characteristics of the 0-1 loss require the use of convex surrogate losses for effective computation (Zhang, 2004; Bartlett et al., 2006). This need is crucial for advancing novel learning algorithms derived from PAC-Bayes bounds. Several studies have addressed this by integrating convex surrogate losses, such as Dalalyan and Tsybakov (2012a) and Alquier et al. (2016), which concentrate on risk bounds for the convexified loss. Despite recent progress in applying PAC-Bayesian techniques to establish prediction bounds in classification, a significant gap remains in providing misclassification risk bounds (Cottet and Alquier, 2018; Mai, 2023; 2024). This paper seeks to fill this gap by focusing on misclassification excess risk bounds for PAC-Bayesian approaches in classification that utilizing a convex surrogate loss.

In this work, we provide a unified procedure to obtain misclassification excess risk bounds for PAC-Bayesian approaches in classification when using convexified loss. We does not present a new result. Fundamentally, the procedure arise from a combination of previous findings in Bartlett et al.

(2006) and Alquier (2024). However, recognizing this connection is crucial for future research aiming to analyze misclassification excess risk bounds for learning methods derived from PAC-Bayesian principles.

We formally consider the following general binary classification. Given a covariate/feature $x \in \mathcal{X}$, one has that the class label $Y = 1$ with probability $p(x)$, and $Y = -1$ with probability $1 - p(x)$, here $p(x)$ denotes the conditional probability $\mathbb{P}[Y = 1|X = x]$. The accuracy of a classifier $\eta$ is defined by the prediction or misclassification error, given as

$$R_{0/1}(\eta) = \mathbb{P}(Y \neq \eta(x)).$$

The Bayes classifier, $\eta^*(x) = \text{sign}(p(x) - 1/2)$, is widely recognized for minimizing $R_{0/1}(\eta)$ (Vapnik, 1998; Devroye et al., 1996), i.e.

$$R^*_{0/1} := R_{0/1}(\eta^*) = \inf R_{0/1}(\eta).$$

With $p(x)$ being unknown, a classifier $\hat{\eta}(x)$ needs to be designed using the available data: a random sample of $n$ independent observations $D_n = \{(x_1, y_1), \ldots, (x_n, y_n)\}$. The design points $x_i$ may be considered as fixed or random. The corresponding (conditional) prediction error of $\hat{\eta}$ is now as

$$R_{0/1}(\hat{\eta}|D_n) = \mathbb{P}(Y \neq \hat{\eta}(x) \mid D_n)$$

and the goodness of $\hat{\eta}$ with respect to $\eta^*$ is measured by the *misclassification excess risk* (Abramovich and Grinshtein, 2018), defined as

$$\mathbb{E}\, R_{0/1}(\hat{\eta}|D_n) - R^*_{0/1} = \mathbb{E}\, R_{0/1}(\hat{\eta}|D_n) - R_{0/1}(\eta^*).$$

The empirical risk minimization method is a general nonparametric approach to determine a classifier $\hat{\eta}$ from data, where the true prediction error $R_{0/1}(\eta)$ minimization is replaced by the minimization of the empirical risk $r_n^{0/1}$ over a specified class of classifiers, $\{\eta_\theta : \mathcal{X} \to \{-1, 1\}, \theta \in \Theta\}$, where $r_n^{0/1}$ is given by:

$$r_n^{0/1}(\theta) = \frac{1}{n} \sum_{i=1}^{n} \mathbb{1}\{y_i \neq \eta_\theta(x_i)\}.$$

PAC-Bayesian approaches for binary classification using the 0-1 loss was thoroughly examined in a series of works by Olivier Catoni over 20 years ago, in Catoni (2003; 2004; 2007). However, due to the computational challenges posed by the non-convexity of the zero-one loss function, particularly when dealing with huge and/or high-dimensional data, a convex surrogate loss is often preferred to simplify the computational problem. The convex surrogate loss in PAC-Bayesian approach for classification has been considered in various studies. For example, Alquier et al. (2016) explored a variational inference approach for PAC-Bayesian methods, emphasizing the importance of convexified loss, while Dalalyan and Tsybakov (2012a) and Mai (2024) investigated a PAC-Bayesian method for classification using convex surrogate loss and gradient-based sampling methods such as Langevin Monte Carlo. PAC-Bayesian bounds as in Alquier (2024), when using a convexified loss, often leads to prediction bounds or excess risk with respect to the convexified loss.

Our work is carried out under the so-called low-noise condition. The low-noise condition described is a common assumption in the classification literature, as seen in works such as Mammen and Tsybakov (1999); Tsybakov (2004); Bartlett et al. (2006). The main challenge for any classifier typically lies near the decision boundary $\{x : p(x) = 1/2\}$. In this region, accurately predicting

the class label is particularly difficult because the label information is predominantly noisy. Given this, it is reasonable to assume that $p(x)$ is unlikely to be very close to $1/2$.

**Structure of the paper:** in Section 2, we introduce our primary notations and present our main results. In Section 3, we apply our general procedure to two significant applications: high-dimensional sparse classification and 1-bit matrix completion. To the best of our knowledge, the results obtained for these two problems are novel. We conclude our work in Section 4, while all technical proofs are provided in Appendix A.

## 2 Main result

### 2.1 PAC-Bayesian framework

We observe an i.i.d sample $(X_1, Y_1), \ldots, (X_n, Y_n)$, of a random pair $(X, Y)$ taking values in $\mathcal{X} \times \{-1, 1\}$, from the same distribution $\mathbb{P}$. A set of classifiers is chosen by the user: $\{\eta_\theta : \mathcal{X} \to \{-1, 1\}, \theta \in \Theta\}$. For example, one may have $\eta_\theta(x) = \text{sign}(\langle \theta, x \rangle) \in \{-1, 1\}$. In this paper, the symbol $\mathbb{E}$ will always denote the expectation with respect to the (unknown) law $\mathbb{P}$ of the $(X_i, Y_i)$'s.

Consider a surrogate loss function $\phi : \mathbb{R}^2 \to \mathbb{R}^+$ that is convex with respect to its second component. Put $\phi_i(\theta) := \phi(Y_i, \eta_\theta(X_i))$ and the empirical convex risk is defined as

$$r_n^\phi(\theta) := \frac{1}{n} \sum_{i=1}^n \phi_i(\theta),$$

and its expected risk is given as

$$R^\phi(\theta) = \mathbb{E}[\phi(Y, \eta_\theta(X))].$$

Convex loss functions commonly used in classification include logistic loss and hinge loss. More examples can be found for example in Bartlett et al. (2006).

Let $\mathcal{P}(\Theta)$ denote the set of all probability measures on $\Theta$. We define a prior probability measure $\pi(\cdot)$ on the set $\Theta$. For any $\lambda > 0$, as in the PAC-Bayesian framework Catoni (2007); Alquier (2024), the Gibbs posterior $\hat{\rho}_\lambda^\phi$, with respect to the convex loss $\phi$, is defined by

$$\hat{\rho}_\lambda^\phi(\mathrm{d}\theta) = \frac{\exp[-\lambda r_n^\phi(\theta)]}{\int \exp[-\lambda r_n^\phi] \mathrm{d}\pi} \pi(\mathrm{d}\theta), \tag{1}$$

and our mean estimator is defined by $\hat{\theta} = \int \theta \hat{\rho}_\lambda^\phi(\mathrm{d}\theta)$. From now, we will let $\theta^*$ denote a minimizer of $R^\phi$ when it exists: $R^\phi(\theta^*) = \min_{\theta \in \Theta} R^\phi(\theta)$.

In PAC-Bayes theory, when utilizing a $\phi$-loss function, it is customary to regulate the excess $\phi$-risk,

$$R^\phi(\theta) - R^\phi(\theta^*)$$

see e.g. Alquier (2024). However, in classification tasks, it is equally crucial to control the misclassification excess risk, $\mathbb{E} R_{0/1}(\theta) - R_{0/1}^*$, which is the primary focus of this paper.

### 2.2 Main result

#### 2.2.1 Assumptions

Certain conditions are essential for deriving our main result.

**Assumption 1** (Bounded loss). *The convex surrogate loss function $\phi$ is assumed to be bounded, with its values lying in the range $[0, B]$.*

The boundedness condition in Assumption 1 is not central to our analysis; rather, it serves to simplify the presentation and enhance the clarity of the paper. It is important to note that PAC-Bayesian bounds can also be derived for unbounded loss functions, as discussed in Alquier (2024).

**Assumption 2** (Lipschitz loss). *We assume that the loss function $\phi(y, \cdot)$ is $L$-Lipschitz in the sense that there exist some constant $L > 0$ such that $|\phi(y, \eta_\theta(x)) - \phi(y, \eta_{\theta'}(x))| \leq L\|\theta - \theta'\|$.*

**Assumption 3** (Bernstein condition). *Assuming that there is a constant $K > 0$ such that, for any $\theta \in \Theta$, $\|\theta - \theta^*\|_2^2 \leq K[R^\phi(\theta) - R^\phi(\theta^*)]$.*

Assumption 2 and 3 have been extensively studied in various forms in the learning theory literature, such as (Mendelson, 2008; Zhang, 2004; Alquier et al., 2019; Elsener and van de Geer, 2018; Alaya and Klopp, 2019). The hinge loss $\phi(y, y') = \max(0, 1 - yy')$ and the logistic loss $\phi(y, y') = \log(1 + \exp(-yy'))$ are 1-Lipschitz with respect to their second argument. Therefore, under appropriate conditions, they satisfy the requirements of Assumption 2 (refer to Section 3 for an example). Assumption 3 implicitly means that our predictors are identifiability.

**Assumption 4** (Definition 4.1 in Alquier (2024)). *Assuming that there is a constant $C_{Berns} > 0$ such that, for any $\theta \in \Theta$, $\mathbb{E}|\phi_i(\theta) - \phi_i(\theta^*)|^2 \leq C_{Berns}[R^\phi(\theta) - R^\phi(\theta^*)]$.*

In fact, our Assumption 3 is more stringent than those presented in Alquier (2024), as formally stated in the following lemma.

**Lemma 1.** *Suppose Assumptions 2 and 3 hold then Assumption 4 is satisfied.*

**Assumption 5** (Margin condition). *We assume that there exist a constant $c > 0$ such that*

$$\mathbb{P}\{0 < |p(X) - 1/2| < 1/(2c)\} = 0.$$

The low-noise condition described in Assumption 5 is a common assumption in the classification literature, as seen in works such as (Abramovich and Grinshtein, 2018; Tsybakov, 2004; Mammen and Tsybakov, 1999; Bartlett et al., 2006). The main challenge for any classifier typically lies near the decision boundary $\{x : p(x) = 1/2\}$, which in logistic regression corresponds to the hyperplane $\theta^\top x = 0$, where $p(x) = (1 + e^{-\theta^\top x})^{-1}$. In this region, accurately predicting the class label is particularly difficult because the label information is predominantly noisy. Given this, it is reasonable to assume that $p(x)$ is unlikely to be very close to $1/2$.

**Assumption 6** (classification-calibrated loss). *For $\zeta \in [0, 1], \zeta \neq 1/2$, the following condition must hold: $\inf_{\alpha \in \mathbb{R}} G_\zeta(\alpha) < \inf_{\alpha : \alpha(2\zeta - 1) \leq 0} G_\zeta(\alpha)$, where $G_\zeta(\alpha) = \zeta\phi(\alpha) + (1 - \zeta)\phi(-\alpha)$.*

Assumption 6 is a minimal requirement, indicating that the $\phi$-loss function possesses the same capacity for classification as the Bayes classifier. For a more detailed discussion, refer to Bartlett et al. (2006).

### 2.2.2 Main results

While high probability PAC-Bayes bounds for the excess $\phi$-risk, $R^\phi(\theta) - R^\phi(\theta^*)$, are frequently discussed in the literature (see e.g. Alquier (2024)), PAC-Bayes bounds in expectation have received comparatively less attention. Utilizing high probability PAC-Bayes bounds for deriving prediction bounds has also been explored to some extent, as evidenced by several works such as Cottet and

Alquier (2018); Mai (2023; 2024). However, these approaches often do not provide bounds for misclassification excess risk unless under strictly noiseless conditions.

In this study, we illustrate the utility of PAC-Bayes bounds in expectation for deriving misclassification excess risk bounds. Specifically, we first introduce a PAC-Bayesian relative bound in expectation, which is a slight extension of Theorem 4.3 in Alquier (2024). For two probability distributions $\mu$ and $\nu$ in $\mathcal{P}(\Theta)$, let $\mathcal{K}(\nu\|\mu)$ denote the Kullback-Leibler divergence from $\nu$ to $\mu$.

Here after, let $\mathbb{P}_n$ and $\mathbb{E}_n$ denote the expectation with respect to the joint distribution of the whole random sample $(X_1, Y_1), \ldots, (X_n, Y_n)$. We first state a main result from Alquier (2024).

**Theorem 1** (Theorem 4.3 in Alquier (2024)). *Assuming that Assumptions 1 and 4 are satisfied, let's take $\lambda = n/\max(2C_{Berns}, C)$. Then we have:*

$$\mathbb{E}_n[\mathbb{E}_{\theta\sim\hat{\rho}_\lambda^\phi}[R^\phi(\theta)]] - R^\phi(\theta^*) \leq 2 \inf_{\rho\in\mathcal{P}(\Theta)} \left\{ \mathbb{E}_{\theta\sim\rho}[R^\phi(\theta)] - R^\phi(\theta^*) + \frac{\max(2C_{Berns}, C)\mathcal{K}(\rho\|\pi)}{n} \right\}.$$

Put $\overline{C} := \max(2L^2K, B)$.

**Corollary 1.** *Assuming that Assumptions 1, 2 and 3 are satisfied, let's take $\lambda = n/\overline{C}$. Then we have:*

$$\mathbb{E}_n[\mathbb{E}_{\theta\sim\hat{\rho}_\lambda^\phi}[R^\phi(\theta)]] - R^\phi(\theta^*) \leq 2 \inf_{\rho\in\mathcal{P}(\Theta)} \left\{ \mathbb{E}_{\theta\sim\rho}[R^\phi(\theta)] - R^\phi(\theta^*) + \frac{\overline{C}\mathcal{K}(\rho\|\pi)}{n} \right\}.$$

The proof is given in Appendix A. As discussed in Catoni (2007); Alquier (2024), the bound in Theorem 1 can be employed to derive error rates for the excess $\phi$-risk in a general setting as follows: one needs to find a $\rho_\epsilon$ such that $\mathbb{E}_{\theta\sim\rho_\epsilon}[R^\phi(\theta)] \simeq R^\phi(\theta^*) + \frac{\epsilon}{n}$ and ensure that $\mathcal{K}(\rho_\epsilon\|\pi) \simeq \epsilon$ to obtain: $\mathbb{E}_n[\mathbb{E}_{\theta\sim\hat{\rho}_\lambda}[R^\phi(\theta)]] \lesssim R^\phi(\theta^*) + \frac{\epsilon}{n} + \frac{2\overline{C}\epsilon}{n}$. Hence the rate is of order $1/n$.

**Remark 1.** *One can derive a PAC-Bayesian relative bound without invoking the Bernstein condition from Assumption 3, see e.g Alquier (2024). Nevertheless, this results in a slower convergence rate of order $n^{-1/2}$. In contrast, under the low-noise condition specified in Assumption 5, which is our primary assumption, it is well-known that a faster rate of order $1/n$ can be obtained Abramovich and Grinshtein (2018); Tsybakov (2004). Hence, the need for imposing the Bernstein condition in Assumption 3 becomes crucial.*

The following theorem presents our main results on misclassification excess risk bounds for PAC-Bayesian approaches in classification using convexified loss. The strategy involves utilizing a broad result from Bartlett et al. (2006). To establish our main result presented in Theorem 2 below, we further assume that the $\phi$-loss function is *classification-calibrated*.

**Theorem 2.** *Assuming both Theorem 1 and Assumption 5, 6 hold, and by selecting $\lambda = n/\overline{C}$, there exists a constant $\Psi > 0$ such that*

$$\mathbb{E}_n[\mathbb{E}_{\theta\sim\hat{\rho}_\lambda^\phi}[R_{0/1}(\theta)]] - R_{0/1}^* \leq \Psi \inf_{\rho\in\mathcal{P}(\Theta)} \left\{ \mathbb{E}_{\theta\sim\rho}[R^\phi(\theta)] - R^\phi(\theta^*) + \frac{\overline{C}\mathcal{K}(\rho\|\pi)}{n} \right\}, \tag{2}$$

*and*

$$\mathbb{E}_n[R_{0/1}(\hat{\theta})] - R_{0/1}^* \leq \Psi \inf_{\rho\in\mathcal{P}(\Theta)} \left\{ \mathbb{E}_{\theta\sim\rho}[R^\phi(\theta)] - R^\phi(\theta^*) + \frac{\overline{C}\mathcal{K}(\rho\|\pi)}{n} \right\}, \tag{3}$$

*in particular, one can take $\Psi = 4c$.*

**Remark 2.** *We emphasize that while the results in Theorem 2 are of significant interest, they do not constitute a new result. Fundamentally, they arise from a combination of previous findings in Bartlett et al. (2006) and Alquier (2024). However, recognizing this connection is crucial for future research aiming to analyze misclassification excess risk bounds for learning methods derived from PAC-Bayesian principles.*

**Remark 3.** *Similar to Theorem 1, the bound in Theorem 2 can be utilized to derive general misclassification error rates. For instance, since the bound in (2) holds for any $\rho \in \mathcal{P}(\Theta)$, one can specify a distribution $\rho_\delta$ such that $\mathbb{E}_{\theta \sim \rho_\delta}[R^\phi(\theta)] - R^\phi(\theta^*) \lesssim \delta/n$ and that $\mathcal{K}(\rho_\delta \| \pi) \lesssim \delta$ and consequently: $\mathbb{E}_n[\mathbb{E}_{\theta \sim \hat{\rho}_\lambda^\phi}[R_{0/1}(\theta)]] - R_{0/1}^* \lesssim \frac{\delta}{n} + \frac{2C\delta}{n}$, hence the misclassification excess rate can be of the order $1/n$. Some classical examples are given below.*

**Remark 4.** *It is crucial to recognize that, in the absence of Assumption 5, one may not achieve a result analogous to Theorem 2. For instance, as demonstrated by Zhang (2004), for the logistic loss, $\mathbb{E}_n[R_{0/1}(\theta)] - R_{0/1}^* \lesssim (\mathbb{E}_n[R^\phi(\theta)] - R^\phi(\theta^*))^{1/2}$. Consequently, it is generally unlikely to derive a comparable result for PAC-Bayesian methods without employing Assumption 5.*

### 2.2.3 Examples

We now demonstrate that using Theorem 2 can yield bounds on the misclassification excess risk in various scenarios. Further non-trivial applications are discussed in Section 3. The following two examples are similar to Example 2.1 and 2.2 in Alquier (2024), but we focus on binary classification setting.

**Example 1** (Finite case)**.** *Let us begin with the special case where $\Theta$ is a finite set, specifically, $\text{card}(\Theta) = M < +\infty$. In this scenario, the Gibbs posterior $\hat{\rho}_\lambda^\phi$ of (1) is a probability distribution over the finite set $\Theta$ defined by*

$$\hat{\rho}_\lambda^\phi(\theta) = \frac{\mathrm{e}^{-\lambda r_n^\phi(\theta)} \pi(\theta)}{\sum_{\vartheta \in \Theta} \mathrm{e}^{-\lambda r_n^\phi(\vartheta)} \pi(\vartheta)}.$$

*As the bounds in (2) and (3) hold for all $\rho \in \mathcal{P}(\Theta)$, it holds in particular for all $\rho$ in the set of Dirac masses $\{\delta_\theta, \theta \in \Theta\}$. That*

$$\mathbb{E}_n[\mathbb{E}_{\theta \sim \hat{\rho}_\lambda^\phi}[R_{0/1}(\theta)]] - R_{0/1}^* \leq \Psi \inf_{\theta \in \Theta} \left\{ R^\phi(\theta) - R^\phi(\theta^*) + \frac{\overline{C}\mathcal{K}(\rho \| \pi)}{n} \right\},$$

*and in particular, for $\theta = \theta^*$, this becomes*

$$\mathbb{E}_n[\mathbb{E}_{\theta \sim \hat{\rho}_\lambda^\phi}[R_{0/1}(\theta)]] - R_{0/1}^* \leq \Psi \frac{\overline{C}\mathcal{K}(\delta_\theta \| \pi)}{n},$$

*And, $\mathcal{K}(\delta_\theta \| \pi) = \sum_{\theta' \in \Theta} \log \left( \frac{\delta_\theta(\theta')}{\pi(\theta')} \right) \delta_\theta(\theta') = \log \frac{1}{\pi(\theta)}$. This gives us an insight into the role of the measure $\pi$: the bound will be tighter for $\theta$ values where $\pi(\theta)$ is large. However, $\pi$ cannot be large everywhere because it is a probability distribution and that $\sum_{\theta \in \Theta} \pi(\theta) = 1$. The larger the set $\Theta$, the more this total sum of 1 will be spread out, resulting in large values of $\log(1/\pi(\theta))$. If $\pi$ is the uniform probability distribution, then $\log(1/\pi(\theta)) = \log(M)$, and the previous bound becomes*

$$\mathbb{E}_n[\mathbb{E}_{\theta \sim \hat{\rho}_\lambda^\phi}[R_{0/1}(\theta)]] - R_{0/1}^* \leq \Psi \overline{C} \frac{\log(M)}{n}.$$

*Thus, in this case, the misclassification excess risk is of order $\log(M)/n$.*

**Example 2.** *Now, we consider the continuous case where $\Theta = \mathbb{R}^d$, the loss function is Lipschitz, and the prior $\pi$ is a centered Gaussian: $\mathcal{N}(0, \sigma^2 I_d)$, where $I_d$ denotes the $d \times d$ identity matrix. When applying Theorem [2], the right-hand side in [(2)] involves an infimum over all $\rho \in \mathcal{P}(\Theta)$. However, for simplicity and practicality, it is advantageous to consider Gaussian distributions as $\rho = \rho_{m,s} = \mathcal{N}(m, s^2 I_d)$ with $m \in \mathbb{R}^d, s > 0$.*

*First, it is well known that, $\mathcal{K}(\rho_{m,s} \| \pi) = \frac{\|m\|^2}{2\sigma^2} + \frac{d}{2} \left[ \frac{s^2}{\sigma^2} + \log(\frac{\sigma^2}{s^2}) - 1 \right]$. Moreover, the risk $R^\phi$ inherits the Lipschitz property of the loss, that is, for any $(\theta, \vartheta) \in \Theta^2$, $R^\phi(\theta) - R^\phi(\vartheta) \le L\|\vartheta - \theta\|$. And, by Jensen's inequality, that $\mathbb{E}_{\theta \sim \rho_{m,s}} \|\vartheta - \theta\| \le \sqrt{\mathbb{E}_{\theta \sim \rho_{m,s}} [\|\theta - m\|^2]} \le s\sqrt{d}$. Consequently, putting all thing together, with $m = \theta^*$*

$$\mathbb{E}_n[\mathbb{E}_{\theta \sim \hat{\rho}_\lambda^\phi}[R_{0/1}(\theta)]] - R_{0/1}^* \le \Psi \inf_{s > 0} \left\{ Ls\sqrt{d} + \overline{C} \frac{\frac{\|\theta^*\|^2}{2\sigma^2} + \frac{d}{2} \left[ \frac{s^2}{\sigma^2} + \log(\frac{\sigma^2}{s^2}) - 1 \right]}{n} \right\}.$$

*Taking $s = 1/(n\sqrt{d})$,*

$$\mathbb{E}_n[\mathbb{E}_{\theta \sim \hat{\rho}_\lambda^\phi}[R_{0/1}(\theta)]] - R_{0/1}^* \le \Psi \left\{ \frac{L}{n} + \overline{C} \frac{\frac{\|\theta^*\|^2}{2\sigma^2} + \frac{d}{2} \left[ \frac{1}{n^2 d\sigma^2} + \log(n^2 d\sigma^2) - 1 \right]}{n} \right\} \lesssim \frac{d\log(n)}{n}.$$

*Thus, in this case, the misclassification excess risk is of order $d\log(n)/n$.*

## 3 Application

We note that Theorem [2] is applicable to different classification contexts. Here, we will demonstrate it with the following two important examples.

### 3.1 High dimensional sparse classifcation

In this context, we have that $\mathcal{X} = \mathbb{R}^d$ and that $d > n$. Consider the class of linear classifiers, the empirical risk is now given by: $r_n^{0/1}(\theta) = \frac{1}{n} \sum_{i=1}^n \mathbb{1}\{Y_i(\theta^\top X_i) < 0\}$, and the prediction risk $R_{0/1}(\theta) = \mathbb{E}_n \left[ r_n^{0/1}(\theta) \right]$. For the sake of simplicity, we put $R^* := R(\theta^*)$, where $\theta^*$ is the ideal Bayes classifier.

Our analysis is centered on a sparse setting, where we assume $s^* < n$, with $s^* = \|\theta^*\|_0$, denoting the number of nonzero elements in the parameter vector. Here, we primarily focus on the hinge loss, which results in the following hinge empirical risk:

$$r_n^h(\theta) = \frac{1}{n} \sum_{i=1}^n (1 - Y_i(\theta^\top x_i))_+,$$

where $(a)_+ := \max(a, 0), \forall a \in \mathbb{R}$. We consider the following Gibbs-posterior distribution: $\hat{\rho}_\lambda^h(\theta) \propto \exp[-\lambda r_n^h(\theta)]\pi(\theta)$ where $\lambda > 0$ is a tuning parameter and $\pi(\theta)$ is a prior distribution, given in [(4)], that promotes (approximately) sparsity on the parameter vector $\theta$. Given a positive number $C_1$, for all $\theta \in \mathcal{B}_1(C_1) := \{\theta \in \mathbb{R}^d : \|\theta\|_1 \le C_1\}$, we consider the following prior,

$$\pi(\theta) \propto \prod_{i=1}^d (\tau^2 + \theta_i^2)^{-2}, \tag{4}$$

where $\tau > 0$ is a tuning parameter. For technical reason, we assume that $C_1 > 2d\tau$. This prior is known as a scaled Student distribution with 3 degree of freedom. This type of prior has been previously examined in the different sparse problems (Dalalyan and Tsybakov, 2012a;b; Mai, 2024).

**Theorem 3.** *Given that $\mathbb{E}\|X\| \leq C_{\mathrm{x}} < \infty$, Theorem 1 and Assumption 5 are satisfied, and by setting $\lambda = n/\overline{C}$, it follows that*

$$\mathbb{E}_n[\mathbb{E}_{\theta \sim \hat{\rho}_\lambda^h}[R_{0/1}(\theta)]] - R_{0/1}^* \leq C \frac{s^* \log(d/s^*)}{n},$$

*and*

$$\mathbb{E}_n[R_{0/1}(\hat{\theta})] - R_{0/1}^* \leq C \frac{s^* \log(d/s^*)}{n},$$

*for some universal constant $C > 0$ depending only on $K, B, C_1, C_{\mathrm{x}}$.*

**Remark 5.** *According to Theorem 3, the misclassification excess rate is of order $s^* \log(d/s^*)/n$ which is established as minimax-optimal in high-dimensional sparse classification, according to Abramovich and Grinshtein (2018). This result is novel and extends the work of Mai (2024), which addresses only the misclassification excess rate in the noiseless scenario.*

### 3.2 1-bit matrix completion

For sake of simplicity, for any positive integer $m$, let $[m]$ denote $\{1, \ldots, m\}$.

Formally, the 1-bit matrix completion problem can be defined as a classification problem as follow: we observe $(X_k, Y_k)_{k \in [n]}$ that are $n$ i.i.d pairs from a distribution $\mathbb{P}$. The $X_k$'s take values in $\mathcal{X} = [d_1] \times [d_2]$ and the $Y_k$'s take values in $\{-1, +1\}$. Hence, the $k$-th observation of an entry of the matrix is $Y_k$ and the corresponding position in the matrix is provided by $X_k = (i_k, j_k)$.

Here, a predictor is a function $[d_1] \times [d_2] \to \mathbb{R}$, and it can therefore be represented by a matrix $M$. A natural approach is to employ $M$ such that when $(X, Y) \sim \mathbb{P}$, the predictor $M$ predicts $Y$ using $\mathrm{sign}(M_X)$. The performance of this predictor in predicting a new matrix entry is subsequently measured by the risk

$$R(M) = \mathbb{E}_{\mathbb{P}}[\mathbb{1}(YM_X < 0)],$$

and its empirical counterpart is: $r_n(M) = \frac{1}{n} \sum_{k=1}^n \mathbb{1}(Y_k M_{X_k} < 0) = \frac{1}{n} \sum_{k=1}^n \mathbb{1}(Y_k M_{i_k, j_k} < 0)$.
From the classification theory (Vapnik, 1998), the best possible classifier is the Bayes classifier

$$\eta(x) = \mathbb{E}(Y|X = x) \quad \text{or equivalently} \quad \eta(i, j) = \mathbb{E}[Y|X = (i, j)],$$

and equivalently we have a corresponding optimal matrix $M_{ij}^* = \mathrm{sign}[\eta(i, j)]$. We define $\overline{r_n} = r_n(M^*)$. Note that, clearly, if two matrices $M^1$ and $M^2$ are such as, for every $(i, j)$, $\mathrm{sign}(M_{ij}^1) = \mathrm{sign}(M_{ij}^2)$ then $R(M^1) = R(M^2)$, and obviously, $\forall M, \forall (i, j) \in [d_1] \times [d_2], \mathrm{sign}(M_{ij}) = M_{ij}^* \Rightarrow r_n(M) = \overline{r_n}$.

In the paper (Cottet and Alquier, 2018), the authors deal with the hinge loss, which leads to the following so-called hinge risk and hinge empirical risk:

$$R^h(M) = \mathbb{E}_{\mathbb{P}}[(1 - YM_X)_+], \quad r_n^h(M) = \frac{1}{n} \sum_{k=1}^n (1 - Y_k M_{X_k})_+.$$

Specifically, with $M = LR^\top$ and for some large enough $K$ (e.g. $K = \min(d_1, d_2)$), Cottet and Alquier (2018) define the prior distribution as the following hierarchical model:

$$\forall k \in [K], \quad \gamma_k \overset{iid}{\sim} \pi^\gamma,$$

$$L_{i,\cdot}, R_{j,\cdot}|\gamma \overset{iid}{\sim} \mathcal{N}(0, \mathrm{diag}(\gamma)), \forall (i,j) \in [m_1] \times [m_2],$$

where the prior distribution on the variances $\pi^\gamma$ is either the Gamma or the inverse-Gamma distribution: $\pi^\gamma = \Gamma(\alpha, \beta)$, or $\pi^\gamma = \Gamma^{-1}(\alpha, \beta)$.

Let $\theta$ denote the parameter $\theta = (L, R, \gamma)$. As in PAC-Bayes theory Catoni (2007), the Gibbs-posterior is as follows:

$$\widehat{\rho}_\lambda^h(d\theta) = \frac{\exp[-\lambda r_n^h(LR^\top)]}{\int \exp[-\lambda r_n^h]d\pi}\pi(d\theta)$$

where $\lambda > 0$ is a parameter to be fixed by the user.

The paper (Cottet and Alquier, 2018) explores a Variational Bayes (VB) approximation, which facilitates the replacement of MCMC methods with more efficient optimization algorithms. They define a VB approximation as $\widetilde{\rho}_\lambda = \arg\min_{\rho \in \mathcal{F}} \mathcal{K}(\rho \| \widehat{\rho}_\lambda^h)$.

We define $\mathcal{M}(r, B)$ for $r \geq 1$ and $B > 0$ as the set of pairs of matrices $(\bar{U}, \bar{V})$, with dimensions $d_1 \times K$ and $d_2 \times K$ respectively, that meet the conditions $\|\bar{U}\|_\infty \leq B$, $\|\bar{V}\|_\infty \leq B$, $\bar{U}_{i,\ell} = 0$ for $i > r$, and $\bar{V}_{j,\ell} = 0$ for $j > r$. Consistent with Cottet and Alquier (2018); Alquier and Ridgway (2020), we assume that $M^* = \bar{U}\bar{V}^t$ for some $(\bar{U}, \bar{V})$ in $\mathcal{M}(r, B)$.

**Theorem 4.** *Assuming that Theorem 1 and Assumption 5 holds and taking $\lambda = n/\overline{C}$, then we have that*

$$\mathbb{E}_n[\mathbb{E}_{\theta \sim \widetilde{\rho}_\lambda}[R_{0/1}(\theta)]] - R_{0/1}^* \leq C\frac{r(d_1 + d_2)\log(nd_1d_2)}{n},$$

*and*

$$\mathbb{E}_n[R_{0/1}(\hat{\theta})] - R_{0/1}^* \leq C\frac{r(d_1 + d_2)\log(nd_1d_2)}{n},$$

*for some universal constant $C > 0$ depending only on $K, B$.*

**Remark 6.** *The misclassification excess error rate presented in Theorem 4, which is on the order of $r(d_1 + d_2)/n$ (up to a logarithmic factor), is established as minimax-optimal, as demonstrated in Alquier et al. (2019).*

## 4 Concluding discussions

This paper presents misclassification excess risk bounds for PAC-Bayesian approaches in binary classification, achieved through the application of a convex surrogate loss function. The methodology primarily relies on the PAC-Bayesian relative bound in expectation, coupled with the assumption of low noise condition. While our analysis assumes a bounded loss, it is worth mentioning that the findings can be extended to unbounded loss scenarios, given additional conditions as elaborated in Alquier (2024). Once the PAC-Bayesian relative bound in expectation for the chosen loss function is established, our theoretical results are applicable.

In our work, the Bernstein condition is assumed; however, it may not always be necessary. Indeed, as evidenced by several studies Cottet and Alquier (2018); Mai (2024), in the noiseless scenario, the margin condition alone is adequate for deriving a misclassification excess risk bound. Additionally,

Section 6 of Alquier et al. (2019) highlights that, under the hinge loss, the low-noise condition aligns with the Bernstein condition. This suggests that investigating the relationship between the Bernstein condition on convex loss and the margin condition within PAC-Bayes bounds could be a valuable area for future research.

### Acknowledgments

The author wishes to thank the Action Editor and three anonymous reviewers for their insightful feedback, which helped enhance the clarity and quality of the paper.

### Conflicts of interest/Competing interests

The author declares no potential conflict of interests.

## A    Proofs

### A.1    Proof of Section 2

**Proof of Lemma 1.** Under Assumption 2, we have that $\mathbb{E}|\phi_i(\theta) - \phi_i(\theta^*)|^2 \leq L^2\|\theta - \theta^*\|_2^2$. Then, under Assumption 3, we have that $L^2\|\theta - \theta^*\|_2^2 \leq L^2 K[R^\phi(\theta) - R^\phi(\theta^*)]$. Thus, Assumption 4 is satisfied with constant $C_{Berns} = L^2 K$. $\qquad\square$

**Proof os Corollary 1.** As Assumptions 2 and 3 hold, then from Lemma 1 we have the Assumption 4 is satisfied. Therefore, We can apply Theorem 1 with $C_{Berns} = L^2 K$ to obtain the result. $\quad\square$

**Proof of Theorem 2.** As Assumption 5 is satisfied, according to Theorem 3 in Bartlett et al. (2006) (taking $\alpha = 1$ and $\psi(t) = t^2$), one has that

$$R_{0/1}(\theta) - R^*_{0/1} \leq 4c\left[R^\phi(\theta) - R^\phi(\theta^*)\right],$$

integrating with respect to $\hat{\rho}^\phi_\lambda$, and then taking the expectation on both sides of the inequality,

$$\mathbb{E}_n[\mathbb{E}_{\theta \sim \hat{\rho}^\phi_\lambda}[R_{0/1}(\theta)]] - R^*_{0/1} \leq 4c\left(\mathbb{E}_n[\mathbb{E}_{\theta \sim \hat{\rho}^\phi_\lambda}[R^\phi(\theta)]] - R^\phi(\theta^*)\right),$$

we obtain the result in (2) by utilizing the result from Theorem 1.

To obtain (3), as $\phi$ is convex, an application of Jensen's inequality to Theorem 1 yields

$$\mathbb{E}_n[R^\phi(\hat{\theta})] - R^\phi(\theta^*) \leq \mathbb{E}_n\mathbb{E}_{\theta \sim \hat{\rho}^\phi_\lambda}[R^\phi(\theta)] - R^\phi(\theta^*)$$

thus we can now apply Theorem 3 in Bartlett et al. (2006) to get that

$$\mathbb{E}_n[R_{0/1}(\hat{\theta})] - R^*_{0/1} \leq 4c\left(\mathbb{E}_n[R^\phi(\hat{\theta})] - R^\phi(\theta^*)\right),$$

and the result is followed. This completes the proof. $\qquad\square$

### A.2    Proof of Section 3

**Proof of Theorem 3.** As the hinge loss is 1-Lipschitz, one has that

$$R^\phi(\theta) - R^\phi(\theta^*) \leq \mathbb{E}\|X\|\|\theta - \theta^*\|$$

We define the following distribution as a translation of the prior $\pi$,

$$p_0(\beta) \propto \pi(\beta - \beta^*)\mathbb{1}_{\mathcal{B}_1(2d\tau)}(\beta - \beta^*). \tag{5}$$

From Lemma 2, we have, for $\rho := p_0$, that

$$\int [R^\phi(\theta) - R^\phi(\theta^*)]p_0(d\theta) \leq C_{\mathrm{x}} \int \|\beta - \beta^*\|p_0(d\beta) \leq C_{\mathrm{x}} \left( \int \|\beta - \beta^*\|^2 p_0(d\beta) \right)^{1/2} \leq C_{\mathrm{x}}\sqrt{4d\tau^2}$$

and

$$\mathcal{K}(p_0\|\pi) \leq 4s^* \log\left(\frac{C_1}{\tau s^*}\right) + \log(2).$$

Plug-in these bounds into inequality (2), one gets that

$$\mathbb{E}_n[\mathbb{E}_{\theta\sim\hat{\rho}_\lambda^\phi}[R_{0/1}(\theta)]] - R_{0/1}^* \leq \Psi \inf_{\tau\in(0,C_1/2d)} \left\{ C_{\mathrm{x}}2\tau\sqrt{d} + \frac{C_1 4s^* \log\left(\frac{C_1}{\tau s^*}\right) + \log(2)}{n} \right\},$$

and the choice $\tau = (C_{\mathrm{x}} n\sqrt{d})^{-1}$ leads to

$$\mathbb{E}_n[\mathbb{E}_{\theta\sim\hat{\rho}_\lambda^\phi}[R_{0/1}(\theta)]] - R_{0/1}^* \leq \Psi \left\{ \frac{2}{n} + \frac{C_1 4s^* \log\left(\frac{C_{\mathrm{x}} C_1 n\sqrt{d}}{s^*}\right) + \log(2)}{n} \right\} \leq c\frac{s^* \log(d/s^*)}{n},$$

for some positive constant $c$ depending only on $L, K, B, C_1, C_{\mathrm{x}}$. A similar argument application to inequality (3), one gets that

$$\mathbb{E}_n[R_{0/1}(\hat{\theta})] - R_{0/1}^* \lesssim \frac{s^* \log(d/s^*)}{n}.$$

The proof is completed.  $\square$

**Proof of Theorem 4.** Using similar argument as in the proof of Theorem 4.3 in Alquier (2024) (see also the proof of Theorem 4.3 in Alquier et al. (2016)), one obtains that

$$\mathbb{E}_n[\mathbb{E}_{\theta\sim\widetilde{\rho}_\lambda}[R^\phi(\theta)]] - R^\phi(\theta^*) \leq 2 \inf_{\rho\in\mathcal{F}} \left\{ \mathbb{E}_{\theta\sim\rho}[R^\phi(\theta)] - R^\phi(\theta^*) + \frac{C_1\mathcal{K}(\rho\|\pi)}{n} \right\}.$$

A similar argument as in Theorem 2,

$$\mathbb{E}_n[\mathbb{E}_{\theta\sim\widetilde{\rho}_\lambda}[R_{0/1}(\theta)]] - R_{0/1}(\theta^*) \leq \Psi \inf_{\rho\in\mathcal{F}} \left\{ \mathbb{E}_{\theta\sim\rho}[R^\phi(\theta)] - R^\phi(\theta^*) + \frac{C_1\mathcal{K}(\rho\|\pi)}{n} \right\}.$$

As the hinge loss is 1-Lipschitz, and noting that $\theta^* = M^*, \theta = LR^\top$, one has that

$$R^\phi(\theta) - R^\phi(\theta^*) \leq \|\theta - \theta^*\| = \|LR^\top - M^*\|$$

Given $B > 0$ and $r \geq 1$, for any pair $(\bar{U}, \bar{V}) \in \mathcal{M}(r, B)$, we define

$$\rho_n(\mathrm{d}U, \mathrm{d}V, \mathrm{d}\gamma) \propto \mathbf{1}_{(\|U - \bar{U}\|_\infty \leq \delta, \|U - \bar{U}\|_\infty \leq \delta)} \pi(\mathrm{d}U, \mathrm{d}V, \mathrm{d}\gamma), \tag{6}$$

where $\delta \in (0, B)$ to be selected later. For any $(U, V)$ in the support of $\rho_n$, given in (6), one has that

$$
\begin{aligned}
\|M^* - UV^t\|_F &= \|\bar{U}\bar{V}^t - \bar{U}V^t + \bar{U}V^t - UV^t\|_F \\
&\leq \|\bar{U}(\bar{V}^t - V^t)\|_F + \|(\bar{U} - U)V^t\|_F \\
&\leq \|\bar{U}\|_F \|\bar{V} - V\|_F + \|\bar{U} - U\|_F \|V^t\|_F \\
&\leq d_1 d_2 \|\bar{U}\|_\infty^{1/2} \|\bar{V} - V\|_\infty^{1/2} + d_1 d_2 \|V\|_\infty^{1/2} \|\bar{U} - U\|_\infty^{1/2} \\
&\leq d_1 d_2 \delta^{1/2} [B^{1/2} + (B + \delta)^{1/2}] \\
&\leq 2 d_1 d_2 \delta^{1/2} (B + \delta)^{1/2} \leq 2^{3/2} d_1 d_2 \delta^{1/2} B^{1/2}.
\end{aligned}
$$

Thus, with $\delta = B/[8(nd_1 d_2)^2]$, one gets that

$$\mathbb{E}_{\theta \sim \rho_n}[R^\phi(\theta)] - R^\phi(\theta^*) \leq B/n.$$

Now, from Lemma 3 with $\delta = B/[8(nd_1 d_2)^2]$, we have that

$$\frac{1}{n} \mathcal{K}(\rho_n \| \pi) \leq \frac{2(1 + 2a)r(d_1 + d_2) \left[\log(nd_1 d_2) + C_a\right]}{n}.$$

Putting all together,

$$
\begin{aligned}
\mathbb{E}_n[\mathbb{E}_{\theta \sim \widetilde{\rho}_\lambda}[R_{0/1}(\theta)]] - R_{0/1}(\theta^*) &\leq C \left\{ \frac{B}{n} + \frac{2(1 + 2a)r(d_1 + d_2)\left[\log(nd_1 d_2) + C_a\right]}{n} \right\} \\
&\lesssim \frac{r(d_1 + d_2)\log(nd_1 d_2)}{n},
\end{aligned}
$$

for some numerical constant $C > 0$ depending only on $a, C_1$. The proof is completed. □

**Lemma 2.** *Let $p_0$ be the probability measure defined by (5). If $d \geq 2$ then $\int_\Lambda \|\beta - \beta^*\|^2 p_0(d\beta) \leq 4d\tau^2$, and $\mathcal{K}(p_0 \| \pi) \leq 4s^* \log\left(\frac{C_1}{\tau s^*}\right) + \log(2)$.*

*Proof.* See Mai (2024), which utilizes results from Dalalyan and Tsybakov (2012a). □

**Lemma 3.** *Put $C_a := \log(8\sqrt{\pi}\Gamma(a)2^{10a+1}) + 3$ and with $\delta = B/[8(nd_1 d_2)^2]$ that satisfies $0 < \delta < B$, we have for $\rho_n$ in (6) that $\mathcal{K}(\rho_n \| \pi) \leq 2(1 + 2a)r(d_1 + d_2)\left[\log(nd_1 d_2) + C_a\right]$.*

*Proof.* This result can found in the proof of Theorem 4.1 in Alquier and Ridgway (2020). □

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
