# OpenReview forum: "Misclassification excess risk bounds for PAC-Bayesian classification via convexified loss"
_TMLR — Rejected by TMLR_

### Review · Reviewer_quSg · 2024-10-04

**Summary Of Contributions:**

This work attempts to link bounds on the excess $\phi$-risk (for a convex surrogate loss $\phi$) to bounds on the excess risk w.r.t $0/1$ misclassification loss. While the misclassification excess risk is the fundamental quantity of interest, it is non-convex and non-smooth. The approach demonstrated by the authors rely on two key prior results,
1. PAC-Bayesian relative oracle bounds (Theorem 4.3, Alquier (2024)).
- Using this one can derive error rates for general settings by choosing an appropriate auxiliary distribution $\rho \in \mathcal{P}(\theta)$ over parameters .
- These bounds are in expectation over the training data rather than high probability.
2. Link between excess $\phi$-risk and excess $0/1$-misclassification risk for convex $\phi$ and data distributions that satisfy low-noise assumptions. (Theorem 3, Bartlett et. al. (2006)).

Using the above two key results, the authors establish their main result in Theorem 2 which in turn results in novel bounds for two distinct learning tasks - 1 bit matrix completion and high-dimensional sparse classification.

*References*

Alquier, P. (2024). User-friendly introduction to PAC-Bayes bounds. Foundations and Trends® in Machine Learning, 17(2):174–303.

Bartlett, P. L., Jordan, M. I., and McAuliffe, J. D. (2006). Convexity, classification, and risk bounds. Journal of the American Statistical Association, 101(473):138–156

**Audience:**

Yes

**Broader Impact Concerns:**

I have no concerns about the broader impact of this article.

**Claims And Evidence:**

Yes

**Requested Changes:**

This manuscript would benefit from a repeated emphasis on the notation and more internal consistency. Below I raise a few specific writing issues that the authors should address,

1. In some parts of the paper, $C$ is a constant, in other parts $C$ is a function tied to classification calibration. Similarly, $B$ is both a constant in some parts and a ball of norm-bounded points. I’d suggest the authors disambiguate the notation in such instances.

2. There are instances of notation being introduced a couple of lines after they have already been used. For example, K in 1-bit matrix completion is used in the hierarchical specification of the prior distribution but the definition of K is only clear later when the optimal matrix $M*$ is assumed to belong to a specific subset of low-rank matrices with rank K.

3. The risk $R_{0/1}$ has been defined twice in the introduction, where the second definition reflects a conditioning on dataset $D_n$. However it is unclear how this condition impacts the definition of the risk or the chosen classifier $\eta$, i.e. the left hand side in these definitions prescribes the risk of a classifier $\eta$. Once such a classifier is specified, what does it mean to condition on the dataset?
Perhaps the authors intend to convey $\eta = A(D_n)$  for some learning algorithm $A$. If this is the case, then perhaps the left hand side should be written as the risk of the outcome of a learning algorithm once a dataset $D_n$ has been additionally provided. This is a point that needs to be clarified in any revised version of the manuscript.

4. The convex risk $\phi$ has domain $R^2$. As such convexity can refer to convexity with respect both the arguments or one of them. Instead, it appears the authors require the convexity of $\phi$ w.r.t parameters $\theta$? The writing should be amended to clarify this.

5. In remark 1, the function $\phi$ only has one argument, but the definition of the surrogate $\phi$ requires two arguments. Please make the notation consistent.

6. Why is the classification-calibration not an official assumption similar to A1-4 ?


7. The definition of Lipschitzness for the convex surrogate $\phi$ is w.r.t the parameters $\theta$. The authors later state that the hinge-loss and logistic losses are examples of $1$-Lipschitz loss functions but here they refer to Lipschitzness w.r.t the function arguments. Granted these loss functions in concert with a L-Lipschitz set of classifiers lead to L-Lipschitz surrogate loss functions w.r.t parameters.

**Strengths And Weaknesses:**

**Strengths**
Novel bounds are demonstrated for two distinct learning tasks - 1-bit matrix completion and high-dimensional sparse classification.


**Weakness**
- How does the constant $c$ in the margin assumption (Assumption 4) impact the results? The core contribution of this article is a misclassification excess risk bound in the presence of label noise. Hence it would be useful to understand how larger noise (via appropriate choice of c) impacts the quality of the bound.
- There is insufficient discussion on the pros and cons of seeking bounds in expectation vs high probability.

---

### Review · Reviewer_iQPa · 2024-10-06

**Summary Of Contributions:**

As far as I can see, this paper is about binary classification problems. The paper is about excess risk bounds for the probability of error ("misclassification") of some classification rules. The PAC-Bayes bounds, alluded in the title, correspond to the expected risk of randomised classification rules defined by distributions on some parametric class of classifiers. The work claims to present excess risk bounds for such kinds of randomised classifiers, under some assumptions, and for the case of a convex loss function.

**Nota bene:** I have responded "No" to the questions "Claims and Evidence" and "Audience" below. This is mainly to do with my concerns regarding the clarity and readability of the paper (its lack of, more precisely). That said, I am open to changing these answers if a revised paper clears my concerns and objections, as described below in the "Requested Changes" section.

**Audience:**

Yes

**Claims And Evidence:**

No

**Requested Changes:**

- Title: What is the intended meaning of "PAC-Bayesian classification" here?
- Since "PAC-Bayesian classification" isn't a standard definition, its presence in the title needs to be explained at some point. Or else, if there is no strong argument to defend it, perhaps the title needs to be reformulated as per the intended meaning.
- Abstract: Preferably, an abstract should be a concise informative summary of the paper. Informative meaning that it should clearly outline (1) the problem (and/or problem area) the paper targets, (2) how the target problem is approached, and (3) highlights of results. The current abstract needs to be re-written to meet these criteria.

**Section 1**
- The first paragraph says some things that are not completely clear and may lead to confusions. Strictly speaking, the PAC-Bayes bounds do not require a convex loss. The zero/one loss is completely fine in PAC-Bayes bounds. In fact, when using a PAC-Bayes bound for computing a numerical generalisation bound value for a randomised classification model, such as the PAC-Bayes-kl bound, the bound is applied to the probability of error, i.e. risk under the zero-one loss. As far as I know, any bounded loss can be used in the PAC-Bayes bounds this way, boundedness being the strict requirement. Once again, the requirement of convex loss is not necessary for evaluation of the PAC-Bayes bounds. The author(s) may have in mind that for *optimisation* the requirement of convex loss is convenient, and indeed widely used. Perhaps the authors have in mind the convex loss to use PAC-Bayes bounds as learning objectives?
- A bit more problematic is the fact that the motivation for the work is rather weak. The author(s) should make an effort to write this first part of the introduction such as to help readers understand the targeted problem and how the paper proposes to approach it.
- Second paragraph: I object the notation $p(X)$, for several reasons. First of all, make an effort to use notation that distinguishes the label used for a random variable, say $X$ (capital), from the generic possible values it may take, say $x$ (lower case). Already the author(s) used a third different way to denote the space of all possible values $\mathcal{X}$, which is good. Then the notation $p(x)$ is natural for the marginal distribution over inputs, in density form. Which means there is an assumption of some joint distribution for $(X,Y)$, the input-label pair, and $p(x)$ then is the marginal density for the first variable.
- However, the author(s) should use a different notation for clarifying the meaning of "the class label $Y = 1$ with probability $p(X)$, and $Y = −1$ with probability $1 − p(X)$." I think the intended probabilities are $\mathbb{P}[Y=1 | X=x]$ and $\mathbb{P}[Y=-1 | X=x]$. These are sometimes called posterior probabilities, see e.g. [1], which in general depend on the prior probabilities $p_y = \mathbb{P}[Y=y]$, the class-conditional probabilities $p(x|Y=y)$ and the input marginal $p(x)$ via the formula
$$\mathbb{P}[Y=y | X=x] = \frac{p(x|Y=y) p_y}{p(x)}$$
by a simple application of Bayes' rule. Again, see [1], Chapter 2 of that book presents these things clearly. Then, in view of this, a second objection I have is that then setting $\mathbb{P}[Y=1 | X=x] = p(x)$ and $\mathbb{P}[Y=-1 | X=x] = 1-p(x)$ seems to be a special and very strong assumption. Perhaps the authors want to use a different notation, say $\mathbb{P}[Y=1 | X=x] =: \nu(x)$ defining $\nu(x)$ to be this conditional probability, and then of course $\mathbb{P}[Y=-1 | X=x] =: 1-\nu(x)$ since this is the binary classification problem.

[1] Moritz Hardt and Benjamin Recht, PATTERNS, PREDICTIONS, AND ACTIONS.
https://mlstory.org/pdf/patterns.pdf

- On a related note. The author(s) apparently followed Devroye et al. "A Probabilistic Theory of Pattern Recognition" as cited. Why not then follow its notation as well? Of course it is fine if the author(s) preferred notation differs. But make it clear and useful, and consistent throughout the paper. On account of clarity, I think the current $p(X)$ for $\mathbb{P}[Y=1|X]$ is not optimal. In fact, such choice is quite likely to confuse readers (as it has confused me) when trying to understand the paper.
- Page 2, middle of the page: The paragraph starting "PAC-Bayesian classification using [..]" again has some misleading remarks about convexity of the loss function used. Again, the author(s) should clearly distinguish the uses of the PAC-Bayes bounds, whether for bound evaluation or for learning objective, and according to what is the case and how the computational problem is affected by this.
- Next paragraph: The meaning of "PAC-Bayesian classification" needs to be clarified, as per the intended meaning. Also, make sure that the claim "to the best of our knowledge" is fine. I think excess risk bounds for the probability of error of binary classifiers have been studied in the literature before. Perhaps the author(s) should clarify what aspect has not been studied before, as they claim.
- Bottom of page 2, paragraph on organisation of the paper: Try a paragraph title with boldface, to signal, in a way that stands out visually, that this is the description of the organisation of the paper.

**Section 2**
- On a quick run through this section, I am inclined to believe that this paper is an exposition of things that have been seen before in the literature. Please correct me if this is not the case. And in case this is an exposition, I'd be in favour if it provides helpful information and discussions and maybe new examples (I think this paper gives this, the examples) such as to contribute positive value to the existing literature. Also, I'd highlight the requirement of a clearly (ideally nicely) written paper.
- Beginning of the section stated that $\mathbb{E}$ is used to denote the expectation with respect to the (unknown) law $\mathbb{P}$ of the $(X_i, Y_i)$’s. Make sure to clarify whether $\mathbb{P}$ and $\mathbb{E}$ are the probability law and expectation for a single random pair $(X_i, Y_i)$. In some bounds (later in the paper) I get the impression that the intended expectation is with respect to the joint distribution of the whole random sample $(X_1, Y_1), \ldots, (X_n, Y_n)$, which would need a different notation. (Perhaps $\mathbb{P}_n$ and $\mathbb{E}_n$ for simplicity?)

**Section 3**
- When reading "our procedure" I wondered what the intended meaning is. Not obvious at all, currently.
- The chosen examples to instantiate bounds are interesting. I would expect to see clear pointers to related literature though. Otherwise it could be assumed that the author(s) would be making claims of originality in these examples? In what respect?
- In general, try to improve the clarity about what this paper does/contributes and what has been done before.

**Section 4**
- Once again, "PAC-Bayesian classification" is unclear. This should be reformulated as per the intended meaning.
- Perhaps more details and discussions for the unbounded loss case? Plus citing the literature.

**Other**
- The acknowledgements section is empty. Either complete, or else remove.
- In the references, in your bib file, make sure to capital-protect the titles, to read e.g. "PAC" (not Pac nor pac).

**Strengths And Weaknesses:**

**Strengths**
- Binary classification problems are always interesting, for several reasons.

**Weaknesses**
- Writing execution appear to have resulted in a poorly written paper, affecting its readability.
- The previous point is mentioned here because this affects the ability of readers to understand the work.
- There is unclarity about the target problem of the paper, and the restrictions (e.g. convexity) and the approach.
- There is unclarity about the contributions of this work, as in what is new versus what has been done before.
- There is unclarity in the mathematical formulations, which could lead to confusions.

---

> ### Comment · Reviewer_iQPa · 2024-10-28
> **Reaction to the rebuttal**
>
> Many thanks for the time spent in the rebuttal, and for responding to my feedback.
>
> I think my comment about $p(x)$ was misunderstood. I objected this notation mainly because it can be interpreted as the marginal distribution on covariate variable $x$, leading to confusion. If this notation is kept, I suggest to insert a clarification note following the place where this notation is declared in the paper; something to the effect that readers should take this as definition of the notation $p(x)$ as it will be used in this paper, i.e. $p(x) := \mathbb{P}[Y=1 | X=x]$, and not to be confused with the marginal distribution for $x$.
>
> Note that the marginal distribution for $x$ is often denoted $p(x)$ in the literature, but in this paper the notation $p(x)$ is used to represent the conditional probability $\mathbb{P}[Y=1 | X=x]$, so the clarification note will alert readers to avoid the confusion.

---

### Review · Reviewer_th9s · 2024-10-07

**Summary Of Contributions:**

PAC-Bayesian theory has often been used to provide generalization bounds with respect to a convex surrogate to a specific loss function. This paper focuses on providing misclassification excess risk bounds in a binary classification setting using a convexified loss. The relevant definitions are provided in section 2.1. To bound the excess risk (with respect to a convex surrogate), the paper uses PAC-Bayesian relative bounds in expectation. More specifically, using a straightforward extension of a result of Alquier (2024). The main results are encapsulated in theorems 1 and 2. It is to be noted that general relative bounds do not provide the same kind of results as direct bounds on the error rate, but they provide a way to reach a faster rate for the relative performance of the estimator within a restricted model provided suitable assumptions are made. The paper relies on two key assumptions: the so-called noise condition, which assumes that p(x) is not likely to be close to 0.5, and an identifiability assumption. The main results (and rates) are first illustrated in both finite and continuous parameter cases with specific assumptions on the prior. They are then further applied to two specific applications: high dimensional sparse linear classification with scaled student distribution as prior, and 1-bit matrix completion.

**Audience:**

Yes

**Broader Impact Concerns:**

The work is theoretical in nature, and thus a broader impact statement is not applicable.

**Claims And Evidence:**

Yes

**Requested Changes:**

See above. It would be good to change the introduction a little bit and make the contributions and terminology (and also the notation later) a bit clearer. Further, at places, it seems to give the appearance of overselling the results e.g. "Although recent research has made strides in using PAC-Bayesian techniques to establish prediction bounds in classification, these efforts have not succeeded in providing misclassification risk bounds" makes it appear there was some technical challenge that had prevented such results. But on looking at the main results of the paper, it doesn't seem to be the case--the main idea relies on a somewhat minor extension of existing results.

Minor comments:
- In the abstract: "However, it typically focus.." -> However, they typically focus
- In the intro. "developing novel learning algorithms in machine learning" --> It would be good to provide references for papers where new learning algorithms have resulted from PAC-Bayes.
- There are occasional grammatical errors which should be corrected.

**Strengths And Weaknesses:**

Strengths:
- The writing of the paper, particularly the technical sections, is mostly clearly and straightforward.
- Unless I have missed something obvious, the results seem technically correct to me.
- The contribution is clean and straightforward, and illustrated via its usage in a number of settings (finite and infinite) and applications (sparse classification and 1-bit matrix completion).

Weaknesses:
These are not necessarily classifiable as "weaknesses", but some points to consider.
- The introduction of the paper seems to be a little confusing. It should make the difference between traditional PAC-Bayesian bounds and the use of relative bounds in expectation more clear and focus on clarifying the main contribution of the paper at the onset. This is not completely clear till the section containing the main results are read. It seems a little too generic the way it is written. It would also be useful to state the reliance on the assumptions to achieve the rates.
- In a sense, as far as I understand, the paper has a limited new contribution. It adapts the results of Alquier 2024, that uses a PAC-Bayesian relative bound in expectation, and uses it to provide bounds on the excess risk. The setup permits faster rates, in exchange for a somewhat stringent low noise condition.

---

### Comment · Action_Editor_SKzy · 2025-01-17
**Some technical questions.**

I have several questions:

1. In the proof of Theorem 1, why is there an expectation on the r.h.s. of the first inequality? It seems that the Remark 1 had it right not to have the expectation on the r.h.s. (I'd suggest defining \phi_i somewhere outside Remark 1 and maybe reminding the reader here.)

2. In the proof of Theorem 2, what is the role of the expectation in the first inequality (the conclusion of applying Thm 3 of Bartlett). I think both sides are nonrandom. The variable \theta is, as far as I'm aware, not quantified over / defined, and so I take this to be a statement that holds for all theta. It would seem that the second line is not so much an application of integration and fubini, but simple taking the expectation E_n E_posterior on both sides of the inequality, since the posterior is not defined without data and so E_posterior E_n never made sense. Right? (In contrast, \hat\theta is a random variable in the proof of (3) and so the expectation at least has a role to play.)

3. In the proof of Theorem 2, when you invoke Theorem 3 of Bartlett et al. (2006), why do you take \psi(t) to be \psi(t) = t^2? As far as I'm aware \psi(t) = \phi(0) - H((1+t)/2) and H is the optimal conditional phi-risk. I don't see why this should be \psi(t)=t^2. Perhaps it doesn't matter since \psi(0) = 0, and, with \alpha=1, the exponent 1-\alpha kills the risk term inside \psi in Theorem 3, but I just didn't understand why you suggest we are taking \psi(t)=t^2.

4. In the proof of Theorem 2, you write "To obtain (3), as φ is convex, an application of Jensen’s inequality to Theorem 1 yields...". While \phi is convex, \theta \mapsto \eta_\theta(x) is not. Can you break down this argument into smaller steps to clarify?

---

> ### Author Response · Authors · 2025-01-21
> **rely to Action Editor SKzy**
>
> Thank you very much for your careful reading and insightful comments.
> ## reply Question 1:
> Thank you. You are right, the expectation on the r.h.s. of the first inequality is not necessary. We have now removed it in the proof and in the Remark 1. Yes, \phi_i has now been clearly stated right after the introduction of the \phi function.
> “{\color{red} Put $ \phi_i (\theta) := \phi(Y_i, \eta_{\theta}(X_i)) $ and}”.
> And, we have also remined the reader its definition in the proof.
>
> ## reply Question 2:
> Yes, you are absolutely right! First, both sides are nonrandom and to be a statement that holds for all \theta. And, thank you very much for correcting this point, we do not need an application of Fubini here. This point is now corrected in to simply taking the expectation E_n E_posterior on both sides of the inequality.
>
> ## reply Question 3:
> Yes, You are right about that function as the optimal one gives a tight bound. But \psi(t) = t^2 is just a valid example so that we can obtain some explicit constant (not the best one). And, actually, you are also absolutely correct since \psi(0) = 0, and, with \alpha=1, the exponent 1-\alpha kills the risk term inside \psi in Theorem 3. The reason taking \psi(t)=t^2 comes from the sentence stated right above the Theorem 3 of Bartlett et al. (2006): “In cases where \psi is strictly convex, such as the exponential, quadratic, and logistic loss functions, this implies that performance improves in the presence of a favorable noise exponent, without knowledge of the noise exponent.” This kind of function had also been used in [Abramovich, F. and Grinshtein, V. (2018). High-dimensional classification by sparse logistic regression. IEEE Transactions on Information Theory, 65(5):3068–3079.] (cited in the manuscript.)
>
> ## reply Question 4:
> Thank you so much for this point. You are right. This is our mistake; \theta \mapsto \eta_\theta(x) is not convex. Our analysis is only true for convex mapping for example: the class of linear classifiers as in our subsection 3.1 that \theta \mapsto \eta_\theta(x):= \theta^\top x. We have now removed all statement related to this mean estimator accordingly.

---

> > ### Comment · Action_Editor_SKzy · 2025-01-21
> > **Further comments.**
> >
> > The paper  needs some further revisions to give readers a clear picture of its contributions, above and beyond the literature.
> >
> > You should introduce a lemma or proposition that shows that Assumptions 2 and 3 give you the assumption of Theorem 4.3 in Alquier. I would maybe introduce this as a new Assumption so you can refer to it.
> >
> > I'd then QUOTE Alquier's theorem (i.e., \begin{theorem}[{\citep[][Thm.~4.3]{alquier}}] ...) and express your result as a corollary.
> >
> > It's clear from the proof that there's nothing new here. Indeed, your result is not an EXTENSION of Alquier's. It is a special case. You've taken Alquier's result which is of the form A ==> B and you have taken two assumptions C,D and shown (C and D) ==> A hence (C and D) ==> B. This is not an extension, logically.  (Such special cases can be useful if C, D are much more readily verified. However, C and D are regularly used in the literature to derive A, and I think we see this even in Bartlett et al.)
> >
> > Theorem 2 is barely a new contribution. It is the composition of the noise condition result of Bartlett and Alquier's theorem. This should be made clearer in the body of the paper.
> >
> > Finally, Corollary 1 is a corollary of Alquier and Assumption 3, whereas its placement now suggests it has something to do with Theorem 2, when it does not have much if anything to do with Theorem 2.
> >
> > With these revisions in place, the abstract and intro need to be substantially revised to more accurately convey the contributions here. If there is a contribution in this paper, it is noticing that one can combine existing PAC-Bayes bounds with Bartlett's result, offering us the analyses you highlight in the examples.

---

> > > ### Author Response · Authors · 2025-02-16
> > > **response to 'Further comments' by Action Editor SKzy**
> > >
> > > Dear Action Editor SKzy,
> > > Thank you so much for your comments.
> > >
> > > Yes, we agree with you that we do not provide a new result, but rather we present a procedure to obtain misclassification excess risk bounds for learning methods derived from PAC-Bayesian bounds. (This have already pointed out by reviewers, we did confirm this).
> > >
> > > However, we believe that our procedure has not been used before and thus current misclassification excess risk bounds for learning methods derived from PAC-Bayesian bounds is only for the noiseless (very strictly) case. Our results allows to obtain better results as illustrated in our examples and applications.
> > > We believe that this is of interest to, at least, the PAC-Bayes community and align well with the focus of TMLR.
> > >
> > > 'You should introduce a lemma or proposition that shows that Assumptions 2 and 3 give you the assumption of Theorem 4.3 in Alquier. I would maybe introduce this as a new Assumption so you can refer to it.' --> This is done. Thank for your suggestion.
> > >
> > > 'Theorem 2 is barely a new contribution. It is the composition of the noise condition result of Bartlett and Alquier's theorem. This should be made clearer in the body of the paper.' --> Thank you. Yes, we have now made this point very clearly stated in the introduction and below Theorem 2 as "We emphasize that while the results in Theorem \ref{thm_main} are of significant interest, they do not constitute a new result. Fundamentally, they arise from a combination of previous findings in \cite{bartlett2006convexity} and \cite{alquier2021user}. However, recognizing this connection is crucial for future research aiming to analyze misclassification excess risk bounds for learning methods derived from PAC-Bayesian principles."
> > >
> > > 'Finally, Corollary 1 is a corollary of Alquier and Assumption 3, whereas its placement now suggests it has something to do with Theorem 2, when it does not have much if anything to do with Theorem 2.' --> Yes, we have now removed this.
> > >
> > > Thank you very to make our manuscript more clear.

---

### Decision · Action_Editor_SKzy · 2025-04-05

**Recommendation:** Reject

**Comment:**

The reviewers generally agreed that the paper is technically sound but lacks meaningful contributions beyond the application of existing results. The primary concerns raised during the review process were related to the paper’s framing, its claims of novelty, and the clarity of its contributions. The authors made several revisions in response to reviewer feedback, addressing technical errors and improving the clarity of their presentation. These revisions, however, further highlighted the paper's dependence on prior work.

A key issue identified during the review was the assumption that convexity of the loss function would suffice to apply Jensen’s inequality in the proof of Theorem 2. Upon closer scrutiny, it became clear that the required convexity condition does not hold for general settings, restricting the validity of the result to convex loss classes, such as those induced by linear classifiers. Neural networks would not meet the criteria, e.g.

Additionally, the reviewers noted that the core results of the paper essentially repackage existing theorems from the literature. The realization that the main theoretical contributions can be derived directly from Alquier’s theorem, combined with Bartlett’s result, indicates that the work does not extend the current state of knowledge in a meaningful way. The suggestion to restructure the paper to clearly present these results as corollaries of known theorems further reinforces this conclusion.

Despite the authors' efforts to clarify and refine their work, the fundamental issue remains: the submission does not provide new theoretical contributions that would justify publication in TMLR. The paper might be better suited for a venue focusing on tutorials or applications rather than theoretical advances.

**Audience:**

While the topic of PAC-Bayesian classification and misclassification excess risk bounds is relevant to TMLR's audience, the specific contributions of this paper are unlikely to generate significant interest. The primary reason is that the results do not offer new insights beyond what is already available in the literature. Readers familiar with PAC-Bayesian methods would likely find the paper's insights to be incremental and not sufficiently novel to warrant publication.

Moreover, the practical applications discussed, such as sparse classification and 1-bit matrix completion, do not introduce substantial methodological advancements. These applications rely on existing theoretical tools without demonstrating broader generalizability or impact. Given TMLR's emphasis on publishing work that provides valuable new perspectives to the machine learning community, the current submission does not sufficiently align with these goals.

**Claims And Evidence:**

The claims made in the submission are not fully supported by clear and convincing evidence. While the theoretical results are correct, they are primarily applications of existing work rather than novel contributions. The paper's main result is the literal composition of a theorem due to Alquier and Bartlett et al.'s noise condition. As such, there's no technical novelty here. That said, noise conditions are not commonly exploited in the PAC Bayes literature (though they have been).

The authors' revisions addressed several technical issues that I raised, such as the unnecessary expectation in the proof of Theorem 1 and an incorrect application of Fubini’s theorem in Theorem 2. However, the correction regarding the convexity of the loss function in Theorem 2 revealed a significant gap in the original analysis, requiring the authors to limit their claims to ones admitting a convex "loss class", such as that of linear classifiers (but certainly not neural networks). As a result, there's actual no nontrivial use of convexity in the entire paper. The paper has basically nothing to say about "convexified losses".

Ultimately, the paper does not offer anything new. Nor is it comprehensive enough to be considered a survey. Its main results follow directly from existing theorems, and the approach is better characterized as a straightforward synthesis of known techniques rather than a contribution that advances the state of the art. The presentation of the work as an extension is misleading, as it primarily applies known results under familiar assumptions rather than introducing new theoretical insights. The revisions necessary to present the work in an accurate light would render this into something that would not meet the bar for TMLR.